# Robust Disentanglement of a Few Factors at a Time using rPU-VAE

**Benjamin Estermann**[*]
Institute of Neuroinformatics, ETH Zurich
besterma@ethz.ch

**Markus Marks**[*]
Institute of Neuroinformatics, ETH Zurich
marksm@ethz.ch

**Mehmet Fatih Yanik**
Institute of Neuroinformatics, ETH Zurich
yanik@ethz.ch

## Abstract

Disentanglement is at the forefront of unsupervised learning, as disentangled representations of data improve generalization, interpretability, and performance in downstream tasks. Current unsupervised approaches remain inapplicable for real-world datasets since they are highly variable in their performance and fail to reach levels of disentanglement of (semi-)supervised approaches. We introduce population-based training (PBT) for improving consistency in training variational autoencoders (VAEs) and demonstrate the validity of this approach in a supervised setting (PBT-VAE). We then use Unsupervised Disentanglement Ranking (UDR) as an unsupervised heuristic to score models in our PBT-VAE training and show how models trained this way tend to consistently disentangle only a subset of the generative factors. Building on top of this observation we introduce the recursive rPU-VAE approach. We train the model until convergence, remove the learned factors from the dataset and reiterate. In doing so, we can label subsets of the dataset with the learned factors and consecutively use these labels to train one model that fully disentangles the whole dataset. With this approach, we show striking improvement in state-of-the-art unsupervised disentanglement performance and robustness across multiple datasets and metrics.

## 1 Introduction

Deep Learning has been highly successful in academia as well as industry over the past years, largely in the supervised domain [1]. Unsupervised learning is becoming increasingly important, as for large amounts of datasets human annotation is either limited or not possible and therefore these datasets remain 'unused'. Disentanglement is a sub-field of representation learning that tries to identify the low-dimensional generative factors of high-dimensional data [2]. Higgins et al. [3] introduced the $\beta$-VAE, a stricter regularization of a variational auto-encoder (VAE) [4, 5], and showed that this model can find generative factors in the *dsprites* dataset [6] without any supervision. Subsequently, the field of disentanglement expanded quickly, as the potential benefits of finding a disentangled representation of data are plentiful, i.e. they allow for more interpretability, cross-domain transfer, robustness, generalizability across machine learning disciplines, as well as improvement of the performance of downstream tasks, such as classification [7, 8, 9, 10, 11, 12, 13].

Locatello et al. [7] showed that fully unsupervised disentanglement might not be possible without any inductive biases. Subsequently, Locatello et al. [14, 15] introduced weak and partial supervision for

---

[*]contributed equally. Code: https://github.com/besterma/robust_disentanglement

training disentanglement models to tackle the aforementioned obstacle. There are no labels available for many real-life applications and for some data, generative factors of interest are hard or impossible for humans to annotate. Recently, Duan et al. [8] defined a new, unsupervised heuristic for evaluating the disentanglement performance of models, based on the assumption that models that disentangle well are more likely to be similar to each other than the ones that do not disentangle [16, 17, 18, 19]. They demonstrate that this Unsupervised Disentanglement Ranking (UDR) correlates well with metrics that rely on previously annotated labels across various models and datasets [8]. Yet, the problem of extreme hyperparameter sensitivity and therefore a lack of performance and robustness in training disentanglement models remains a major challenge in disentanglement representation learning [7].

This paper introduces a systematic way to train models for disentanglement and increase overall performance and robustness. Our contributions are several-fold:

- We introduce Population Based Training (PBT) [20] for variational training to overcome hyperparameter sensitivity and achieve consistently high performing models.

- We demonstrate PBT-VAE training performance in the supervised and semi-supervised case and show how it can beat the state-of-the-art.

- We extend our approach to unsupervised learning using UDR (PBT-U-VAE (UDR)) and describe how this approach leads to a very consistent disentanglement of the factors with the highest variance in image space.

- We show how these factors can be used to label the dataset, and novel factors can be learned by removing the learned ones from the dataset.

- We demonstrate how these learned labels can be used to train a PBT-VAE and call our approach the recursive PBT-U-VAE (UDR) (rPU-VAE).

- We evaluate how the rPU-VAE disentangles different datasets in comparison to the state-of-the-art.

- We show how the performance of the rPU-VAE depends on the number of labels generated during training.

## 2 Background and related work

### 2.1 $\beta$-TCVAE

The Variational Autoencoder (VAE) [5, 4] is a widely adapted deep generative model, particularly in state-of-the-art disentanglement approaches. It consists of an encoder network $q(z|x)$ as well as a decoder network $p(x|z)$. It is trained by maximizing the evidence lower bound (ELBO) (Eq. 1 with $\beta$=1). Here $p(z)$ denotes the prior for the latent variables $z$, which is usually isotropic unit Gaussian. The first ELBO term can be viewed as a negative reconstruction error, while the second term penalizes deviations of the latent code from the prior. Higgins et al. [3] introduced $\beta$ as a hyperparameter in the (ELBO) of a VAE to increase the weight on the penalization term as follows:

$$L_\beta = \frac{1}{N} \sum_{n=1}^{N} (\mathbb{E}_q[\log p(x_n|z)] - \beta \mathrm{KL}(q(z|x_n)||p(z))) \tag{1}$$

The authors showed that this loss modification yields increased disentanglement in the representation of data. Chen et al. [21] further decomposed the penalty term $\mathrm{KL}(q(z|x_n||p(z)))$, into an index-code mutual information, a total correlation and a dimension-wise Kullback–Leibler (KL) term:

$$\mathbb{E}_{p(n)}[\mathrm{KL}q(z|n||p(z))] = \mathrm{KL}(q(z,n)||q(z)p(n)) + \mathrm{KL}(q(z)||\prod_j q(z_j)) + \sum_j \mathrm{KL}(q(z_j)||p(z_j)) \tag{2}$$

where $z_j$ denotes the $j$th dimension of the latent variable. This formulation of VAE, called $\beta$-TCVAE, has been previously shown to perform well on some disentanglement tasks [7]. This is why in this study, we focus on the $\beta$-TCVAE as our base model, but in principle, our framework is model agnostic and could therefore be applied to other state-of-the-art models as well.

## 2.2 Population Based Training

PBT is an optimization algorithm introduced by Jaderberg et al. [20], which can jointly optimize a population of models and their hyperparameters. Compared to grid search or sequential optimization, PBT results in more stable training, faster learning, and higher performance. It can outperform heavily tuned hyperparameter schedules. Its effectiveness has been shown over different domains, specifically in domains prone to hyperparameter sensitivity, for example, deep reinforcement learning [20]. The algorithm consists of a population $\mathcal{P}$, with members $\mathcal{M} \in \mathcal{P}$, each member $\mathcal{M} = (\theta, h, p, t)$, where $\theta$ are the parameters of the model, $h$ the hyperparameters, $p$ the score, and $t$ the current step. Furthermore, there are the functions $step : \theta \leftarrow step(\theta, h, t)$ and $eval : p \leftarrow eval(\theta)$. Before the training starts, each $\mathcal{M} \in \mathcal{P}$ randomly initializes its weights $\theta$ and hyperparameters $h$. First, each $\mathcal{M}$ gets trained for a certain amount of steps using $step$, and its performance gets evaluated using $eval$. It is important to note that $eval$ does not have to be in any relation with the loss function used during $step$, specifically, it does not need to be differentiable. Next, meta-optimization is performed, where the parameters $\theta$ and hyperparameters $h$ are updated given the performance of the entire population. This update is achieved through two functions, which get called independently on each $\mathcal{M}$: $exploit$ allows a member to, given its performance, abandon its solution and focus on a more promising one, $explore$ proposes new possible hyperparameters to allow for better exploration of the hyperparameter space. This cycle continues until either the models converge or the computational budget has been exceeded.

## 2.3 Unsupervised Disentanglement Ranking

Building on top of the probabilistic narrative that disentangled representations are more likely to be similar to each other than entangled ones [16, 17, 18, 19], Duan et al. [22] defined a new unsupervised heuristic for disentanglement model selection. More specifically, their approach is based on two main assumptions. First, disentangled representations are, up to permutation and sign inversion, similar to one another, corresponding to a single plausible disentangled generative process. Second, entangled representations are different from one another, as neural networks usually tend to converge to different hidden representations despite being trained on the same task. Therefore, high representational similarity within a set of models indicates a high likelihood of these models being disentangled. To assess a single model's performance, UDR is computed on a set of models, usually between 5 and 50, with different initial weights but trained using the same hyperparameters. The final score of each model is the median overall pairwise comparison with the other models. Each pairwise comparison is scored according to the similarity of the representations. If this process is repeated over multiple hyperparameters, it is possible to draw conclusions about the fitness of each of these hyperparameter settings and to use that knowledge to guide model selection.

## 2.4 Disentanglement metrics

Several metrics have been introduced to assess different aspects of disentanglement [3, 23, 21, 24, 25, 26]. For a detailed overview, we refer to Supplementary D of [7]. In this study, we focus on MIG [21] and DCI  Disentanglement, since it has been demonstrated that for most datasets [7, 15] most of the disentanglement metric correlate. Moreover, the $\beta$-VAE metric [3] as well as the FactorVAE [23] require the true underlying generative model and are therefore not suited. Further, the SAP score [26] does not seem to correlate well with the other metrics [15]. MIG enforces each ground truth factor to be learned by a single latent variable, but allows one latent variable to learn, and therefore entangle, multiple ground truth factors. DCI  Disentanglement [25] allows us to capture this weakness of MIG, and therefore both metrics give a better picture of disentanglement. Moreover UDR, as well as MIG/DCI Disentanglement, indicate improvement of the representation with respect to downstream task performance since they are both highly correlated to it [8, 7]. We compute MIG as well as DCI Disentanglement as described in [7] using `disentanglement-lib`[1].

## 3  (Semi-)supervised PBT-Disentanglement

Locatello et al. [15] introduced a supervision signal in the loss function during training to aid the training process if a limited number of labels are available. They show how models trained in this

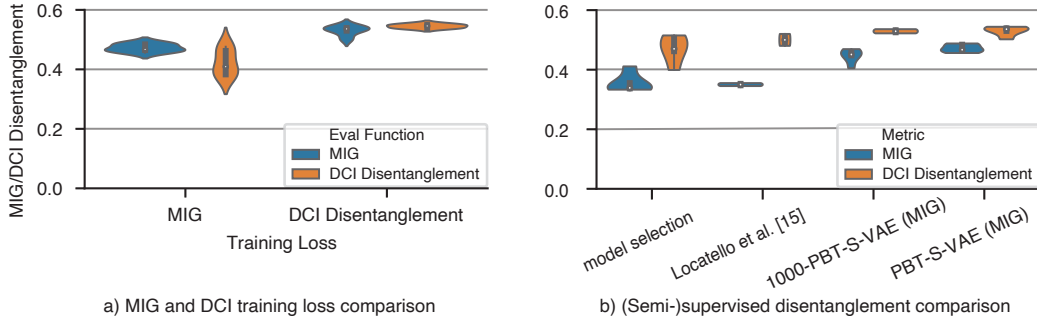

a) MIG and DCI training loss comparison        b) (Semi-)supervised disentanglement comparison

Figure 1: Developing Population Based Training for (semi-)supervised disentanglement. a) MIG is the preferred *eval* function for PBT-S-VAE, as it outperforms DCI Disentanglement when measured with both metrics respectively. b) The semi-supervised 1000-PBT-S-VAE (MIG) outperforms both Locatello et al. [15] and the results of the model selection when all of them are trained on 1000 *dsprites* labels. The fully supervised PBT-S-VAE further increases performance in comparison to the semi-supervised approaches.

semi-supervised fashion outperform unsupervised trained models with respect to disentanglement. We show how our approach of training VAEs using PBT (PBT-VAE) can be used to optimize for disentanglement directly during training. We demonstrate how models trained with PBT for disentanglement express very little variance in disentanglement outcome assessed by MIG and DCI Disentanglement, indicating a robust training process. Moreover, with these supervised PBT-VAE disentanglement results, we establish an upper limit for the realistically achievable unsupervised performance in the subsequent chapter. For evaluation of our approaches in this study, we use the *dsprites* [6] and *shapes3d* [27] datasets, as they are some of the most commonly used in the disentanglement literature and therefore enable us to benchmark against other methods [21, 3, 23, 28, 7, 15, 12].

## 3.1 Comparison of *eval* functions for PBT

PBT-VAE training outcome is contingent on the *eval* function used for PBT. Within our considered metrics of MIG and DCI Disentanglement, we first test what disentanglement each metric used as PBT *eval* function yields, assessed by all other metrics. For each metric, we carry out five separate PBT runs with different random seeds. On *dsprites*, models trained with MIG achieve a higher MIG compared to models trained with DCI Disentanglement, while expressing lower variance (Fig. 1a). Moreover, these models do similarly well, when assessed with DCI Disentanglement compared to models trained with DCI Disentanglement. Based on this experiment, for the rest of the study, we use MIG as *eval* function and call the approach PBT-S-VAE (MIG). Generally, this approach is not limited to either MIG or DCI Disentanglement as an *eval* function and more functions can be assessed in the future for a variety of datasets.

## 3.2 (Semi-)supervised PBT performance

Locatello et al. [15] demonstrated increased disentanglement over the unsupervised baseline using a few-label approach, a realistic alternative to fully supervised approaches for practical applications. We include this approach as a semi-supervised baseline for later comparison. As another baseline, we use the performance of simple model selection. After the finishing training of a set of models, we selected the best model based on their MIG calculated on 1000 randomly selected labels. Finally, we introduce our semi-supervised and fully supervised PBT based approaches. The **semi-supervised** 1000-PBT-S-VAE (MIG) had access a randomly sub-sampled set of 1000 labels, similar to [15] and the model selection baseline, whereas the **supervised** PBT-S-VAE (MIG) was trained using the complete set of labels. All models were trained on the *dsprites* dataset. We see that 1000-PBT-S-VAE (MIG) outperforms both baselines clearly in terms of mean MIG and DCI Disentanglement scores, only surpassed by the PBT-S-VAE (MIG), while showing reduced variance (Fig. 1b).

# 4 Unsupervised PBT-Disentanglement

We now extend our approach to the unsupervised setting after having established the use of PBT for VAE disentanglement training. We use UDR [8] as an unsupervised *eval* function instead of the MIG in the PBT framework, which allows us to move our PBT-S-VAE (MIG) approach towards being an unsupervised method (PBT-U-VAE). We adapt our PBT-VAE approach to fulfill the assumptions of the UDR, namely for PBT-U-VAE (UDR), each $\mathcal{M} \in \mathcal{P}$ consists of five models separately initialized and trained with the same hyperparameters $h$. We denote the weights of all five models as $\theta$. According to Duan et al. [8], five models provide a reasonable estimate of disentanglement, while allowing the approach to be computationally feasible.

## 4.1 Consistent disentanglement of factors with the largest variation

We found that using PBT-U-VAE (UDR), the well-performing models disentangled only ground truth factors with high variance in image space (Fig. 2). While these models did not capture the whole variation in image space, they still achieved high UDR scores and consistently learned the same factors. Once they reached that representation, they failed to capture any new factors of variation during further training epochs. Therefore, the resulting MIG/DCI Disentanglement scores displayed in Fig. 2 are relatively low. This behavior could be accounted for by the fact that the UDR does not consider the number of active latent variables, so a group of models capturing all ground truth factors cannot achieve a better UDR score than models capturing only a distinct subset of those factors.

## 4.2 Recursive PBT-U-VAE (UDR)

We shortly summarize the assumptions we can make given the previous findings:

- PBT-U-VAE (UDR) yields consistent disentangling of a subset of factors.
- PBT-U-VAE (UDR) training converges after these factors have been learned.
- Models, which disentangle those factors, fail to learn any new factors of variation.

These assumptions allow us to design our recursive PBT-U-VAE (UDR) algorithm, overcoming the limitations of the vanilla PBT-U-VAE (UDR) approach. We try to recursively remove varia-

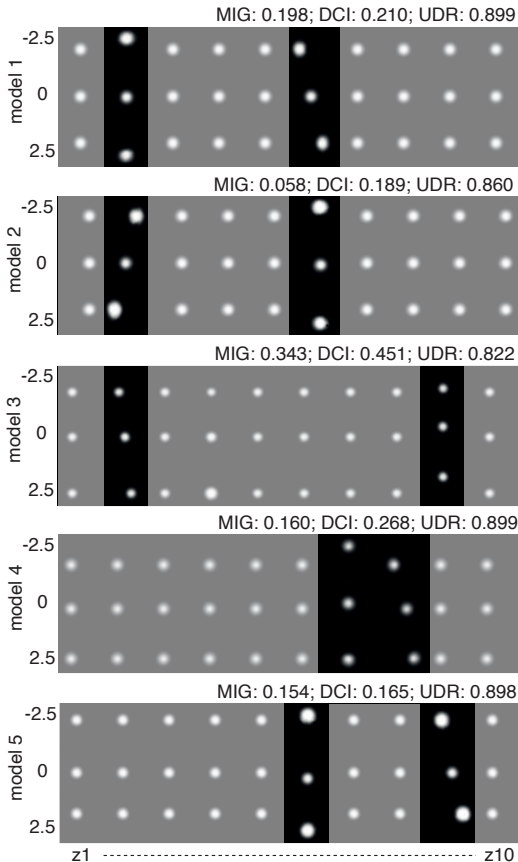

Figure 2: For 5 different random seeds on *dsprites*, PBT-U-VAE (UDR) models consistently disentangle x and y position while reaching high UDR scores. VAE latent variables are indicated with $z1,...,z10$ and active latent variables $z_{active}$ are highlighted by increased contrast.

tion of learned active latent variables from the input dataset. With this reduced dataset, we start another round of PBT-U-VAE (UDR) training, forcing the network to learn new factors of variation. Algorithm 1 shows the implementation of this approach in pseudo-code. For a graphical summary, please see Fig. 7 in the Supplementary A.

**Algorithm 1** recursive PBT-U-VAE (UDR)(rPU-VAE)

```
 1: procedure DISENTANGLE(D)                                              ▷ initial dataset D
 2:     surrogateLabels ← emptyList()
 3:     labels₀ ← PBT-U-VAE (UDR)(D)                                      ▷ Train until convergence
 4:     surrogateLabels.append(labels₀)
 5:     for i ← 1, MaxNrLeafRuns do                                       ▷ Start leaf-runs
 6:         leafLabels ← emptyList()
 7:         d ← reduce(D, labels₀, i)                          ▷ Remove variance of the labeled factor
 8:         while |d| > size_{d-min} ∧ no convergence do       ▷ Recursively label and reduce dataset
 9:             labels_i ← PBT-U-VAE (UDR)(d)                             ▷ This is one metaEpoch
10:             leafLabels.append(labels_i)
11:             d ← reduce(d, labels_i, 0)
12:         end while
13:         surrogateLabels.append(leafLabels)
14:     end for
15:     θ ← PBT-S-VAE (MIG)(D, surrogateLabels)                          ▷ Train final model
16:     return θ
17: end procedure
```

**Initial PBT training until first convergence.** We start with the initial dataset and train our PBT-U-VAE (UDR) approach until the change of UDR score of the best $\mathcal{M} \in \mathcal{P}$ is below a predefined threshold ($\Delta\text{UDR} < \text{threshold}_{\text{UDR}}$) for a fixed number of epochs. If this criterion is reached, we consider this first metaEpoch to be finished. This member $\mathcal{M}$ has now learned a subset of the ground-truth factors.

**Removal of variation explained by learned active latents.** To remove learned factors, we select the best VAE from the set of VAEs of the best member $\mathcal{M} \in \mathcal{P}$. We identify the most active latent variables $z$ of this model by their respective KL divergence to the prior, if $KL(q(z_a|x)||p(z_a)) > threshold_{z_{active}}$, $z_a$ is active, otherwise not, following the argumentation of [8, 19]. We try to reduce the dataset, such that the variance that can be explained by these $z_{active}$ is minimized. To do so, we map the whole dataset onto $z_{active}$ using the encoder $q(z|x)$ of the earlier selected VAE. We use the resulting latent encoding values (LEV) as labels of the dataset, which we refer to as surrogate labels subsequently. Next, we calculate the derivative over the sorted surrogate labels to determine intervals of the dataset over which the selected factors change least (Fig. 3), while trying to keep the resulting dataset as large as possible. To find such intervals, we compute the ratio between the lower of the two adjacent peaks and the mean within the interval.

This ratio allows us to identify intervals which best suit the above mentioned criteria. If multiple active latent variables $z_{active}$ were learned, we intersect the intervals of these factors to get a subset of the dataset where all of these factors vary least. In other words, this is a set intersection of the sets of all elements in the current dataset where the respective latent gets mapped onto an interval of least variation. This reduced dataset will be used for the next metaEpoch of PBT-U-VAE (UDR) training. Having removed most of the variation of the

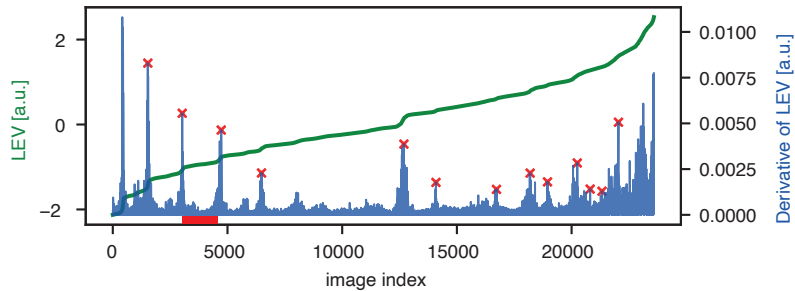

Figure 3: Illustration of the latent encoding and data reduction approach. After one metaEpoch, the best VAE from the population is used to encode the whole *dsprites* dataset onto an $z_{active}$. The data is sorted by their encoding values (green line), and the derivative is computed (blue line). Subsequently, we calculate peaks in the derivatives (red crosses) and select an interval in between the peaks in which the latent variable is not highly variable (red horizontal bar).

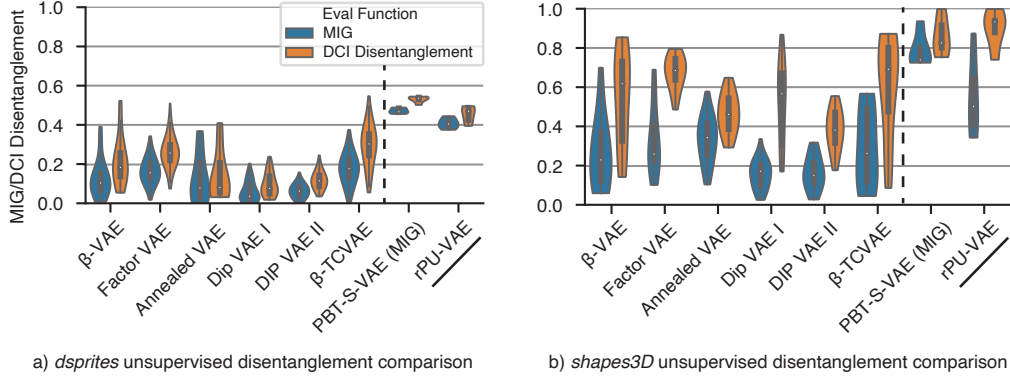

a) *dsprites* unsupervised disentanglement comparison      b) *shapes3D* unsupervised disentanglement comparison

Figure 4: rPU-VAE outperforms previous approaches on commonly used benchmarks. a) comparison of rPU-VAE performance to unsupervised state-of-the-art approaches [21, 3, 23, 26, 28]. The rPU-VAE beats all previous models with respect to mean MIG/DCI Disentanglement scores and has comparatively little variance. It almost reaches disentanglement scores of our supervised PBT-S-VAE (MIG), which we introduced as an upper limit for the performance. b) Same comparison as in a), but on the *shapes3d* dataset. rPU-VAE again outperforms the state-of-the-art and performs very close to our PBT-S-VAE (MIG).

learned latent variables from the dataset forces the network to learn new factors of variation if they exist.

**Leaf-run.** After some number of `metaEpochs` and subsequent dataset reduction, all the variance is removed from the dataset and the models only learn noise and fail to reach a certain UDR score $\forall_{\mathcal{M} \in \mathcal{P}} eval(\theta(M)) < UDR_{threshold}$. Alternatively, we also stop training when the reduced dataset becomes too small ($|d| < size_{d-min}$). One such path of recursive `metaEpochs` we consider as a `leaf-run`. During each `leaf-run`, a subset of the original dataset gets surrogate labels assigned, increasing the number of `leaf-runs`, therefore, increases the amount of total surrogate labels. We could recursively generate a tree from all $z_{active}$ intervals, where the height of the tree is the number of all learned factors, given that the dataset includes all possible combinations of ground truth factors.

**Combining all learned factors into a single model.** With this recursive application of PBT-U-VAE (UDR) and dataset reduction, we can identify most factors of variation of a data distribution and, at the same time, label parts of the data. Subsequently, we use this partially annotated dataset as a supervision signal for the previously mentioned PBT-S-VAE (MIG). The resulting model from PBT-S-VAE (MIG) now captures all of the previously discovered factors at the same time. We name the whole algorithm the recursive PBT-U-VAE (UDR) or rPU-VAE.

## 5 Experiments on unsupervised disentanglement

We demonstrate quantitatively that the rPU-VAE achieves consistently higher scores than state-of-the-art approaches across multiple datasets and metrics while being more robust. We compare the performance of our rPU-VAE approach to the unsupervised $\beta$-TCVAE [21], $\beta$-VAE [3], FactorVAE [23], DIP-VAE-I and DIP-VAE-II [26] and AnnealedVAE [28], where pretrained models are partially available from [7] as part of `disentanglement-lib`. Each of these models was trained with 50 random seeds and a range of hyperparameters, specified in Supplementary E in [7]. Since *shapes3d* models are not publicly available on `disentanglement-lib` we recomputed that data according to [7].

We train our rPU-VAE approach with 5 random seeds and 3 `leaf-runs`. We note that the rPU-VAE achieves significantly higher mean disentanglement scores for all datasets used in Fig. 4. As expected, the robustness of disentanglement is higher (see the drastically reduced variance compared to other approaches). The unsupervised rPU-VAE MIG/DCI Disentanglement scores are very close to our previously introduced supervised PBT-S-VAE (MIG) model. Noticeably, the MIG for *shapes3d* still has a larger gap to the supervised upper-bound when compared to *dsprites*. Models learn color-related ground truth factors with multiple active latent variables, which decreases the MIG for *shapes3d*.

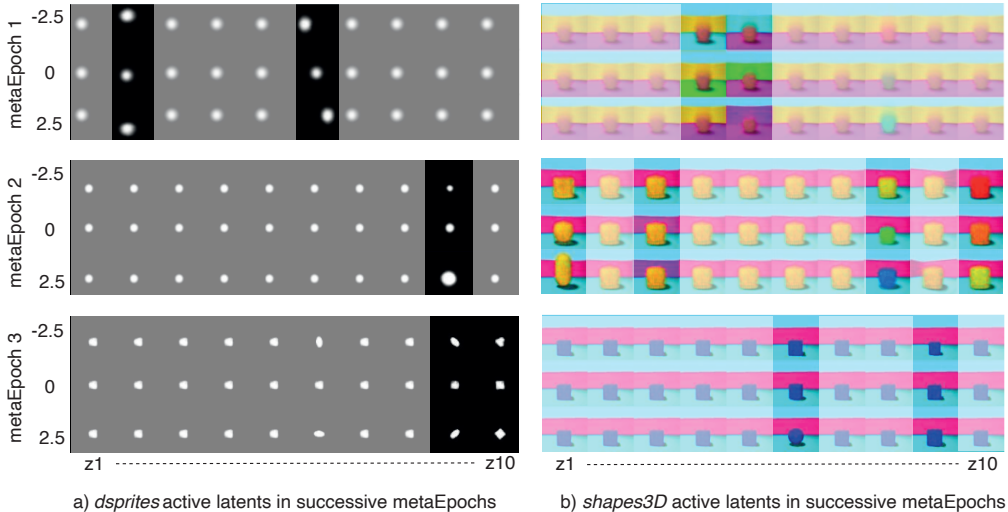

a) *dsprites* active latents in successive metaEpochs

b) *shapes3D* active latents in successive metaEpochs

Figure 5: Representative examples of latent traversals at each of 3 `metaEpochs` during rPU-VAE training. VAE latent variables are indicated with $z1,...,z10$ and active latent variables $z_{active}$ are highlighted by increased contrast. a) Traversals of *dsprites* dataset. `metaEpoch` 1: $z_{active}$ encode y and x position; `metaEpoch` 2: $z_{active}$ code for scale; `metaEpoch` 3: $z_{active}$ code for rotation and shape. b) Traversals of *shapes3d* dataset. `metaEpoch` 1: $z_{active}$ code for wall and floor colors; `metaEpoch` 2: $z_{active}$ code for shape, wall color and twice color of the object; `metaEpoch` 3: $z_{active}$ code for shape of the object and scale.

Nevertheless, we argue that this representation of color is valid and, in fact, disentangled, as each latent variable only represents (parts of) a single ground truth factor, as would one when expressing color using the RGB color model. Looking at what the models learn with each `metaEpoch` (Fig. 5), we can see that at each `metaEpoch` distinct generative factors are learned. We provide more detail on how hyperparameters developed during learning in Supplementary section B while defining all hyperparameters used in Supplementary section E. We show representative full traversals for *shapes3d* in Fig. 12. Moreover, we present full traversals of our models trained on *celebA* [29] and demonstrate the capability of rPU-VAE to find interesting and non-trivial factors in this naturalistic dataset in Fig. 13. Lastly, we include full traversals of the final model on *dsprites* in Fig. 14b. While during `leaf-runs`, our models managed to capture factors such as shape and rotation (Fig. 5a), the final model learns only scale, x-position and y-position. We note that this is the same for the supervised model PBT-S-VAE (Fig. 14a).

## 5.1 Amount of labels needed

As mentioned in Section 4.2, we could potentially aim to label the whole dataset in order to use these surrogate labels for training the rPU-VAE, but this would be computationally expensive. We showed in Section 3.2 that disentangling models can be successfully trained with a fraction of the ground-truth labels, so we assume that this holds for our surrogate labels as well. We test how the number of `leaf-runs` in rPU-VAE training affects MIG/DCI Disentanglement

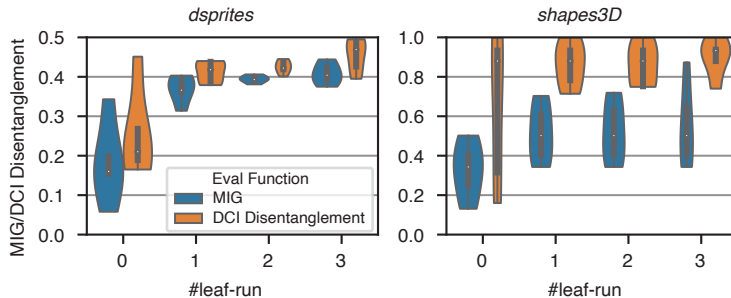

Figure 6: Performance of rPU-VAE with respect to number of `leaf-runs`. For *dsprites* and *shapes3d* MIG/DCI Disentanglement improve with the number of `leaf-runs`, particularly from `leaf-run` 0 to `leaf-run` 1.

performance on *shapes3d* as well as *dsprites*, using 1-3 `leaf-runs` during training for each of the 5 models (Fig. 6). For the *shapes3d* and *dsprites* dataset the performance of rPU-VAE improves over both metrics with the number of `leaf-runs`. We observe the biggest improvement from performing 0 to performing 1 `leaf-run`. Not only does the overall performance improve with `leaf-runs`, but also the variance of the MIG/DCI Disentanglement scores tends to reduce with each `leaf-run`.

## 6   Conclusions

We motivate the rPU-VAE, a new approach for consistent unsupervised disentanglement. We evaluate our approach across multiple datasets and disentanglement metrics against state-of-the-art approaches. The results show the superior performance of rPU-VAE compared to previously used models, not only in terms of improved average performance but also in terms of drastically reduced variance. We experiment with how the number of recursions within the rPU-VAE training affects disentanglement performance. We show how PBT can be used for achieving an increased performance in the semi-supervised setting as well. Finally, we show qualitative results of our method applied to the naturalistic dataset *celebA*. Particularly because of the enhanced robustness, our approach is one step closer to enabling practitioners to leverage the benefits of disentangled representations of their data without the need for annotation.

# 7 Broader Impact

Robust disentangled representations are directly applicable to many domains within science and medicine. In recent years, many unsupervised methods, i.e. dimensionality reduction and manifold techniques, have been used to discover structure in complex datasets in physics [30], genomics [31, 32], neuroscience [33] and energy prediction [34]. Not only do these provide an opportunity in discovering underlying structure in natural phenomena, but also human biases can be reduced by data-driven analysis.

Similarly, there has been a surge in use of unsupervised learning in medicine such as drug design [35], detection of brain lesions [36], cardiac image analysis [37] and drug side effect discovery [38]. Unsupervised approaches have been also highly successful in experimental Brain-Machine-Interfaces (BMI) [39], leading to higher stability of BMI readouts.

However, the enhanced interpretability of disentangled representations in these applications might lead to undue sense of trust and greater negative consequences in case of failures. Thus, further research on reliability of AI approaches has to be done beyond the proof-of-principle.

Our algorithm relies on training large numbers of models, requiring significant computational resources. Therefore energy consumption and resulting $CO_2$ footprint can be quite substantial [40]. The aforementioned benefits of utilizing disentangled representations to improve outcomes in science and medicine are difficult to weigh against the entailing environmental costs and need to be evaluated on a case-by-case basis. Alternatively, further research (such as use of spiking neural networks [41]) may make our approach much more computationally efficient.

# 8 Acknowledgments and Disclosure of Funding

We thank Francesco Locatello for helping us to re-use `disentanglement-lib` and for fruitful discussions. We thank Hafsteinn Einarsson for the very useful feedback on the manuscript.

This project was funded by the Swiss Federal Institute of Technology (ETH) Zurich and the European Research Council (ERC) under the European Union's Horizon 2020 research and innovation program (grant agreement No 818179).

The authors declare no conflicts of interests.

## Footnotes

[1] `https://github.com/google-research/disentanglement_lib`

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
