[Supplementary Material]

# A    rPU-VAE Concept Illustration

Figure 7: Illustration of the rPU-VAE workflow. The initial dataset is used for training a population of UDR-VAE models. After convergence, the best UDR-VAE of the population is selected and an active latent (red unit in yellow VAE encoder) is determined, and the dataset is encoded. Afterwards, labels are sorted, their derivative calculated and a candidate interval selected (red horizontal bar). The dataset is reduced to this interval and PBT-U-VAE (UDR) training is started again. In `leaf-runs`, parts of the dataset are annotated with surrogate labels. After a convergence criterion is met, the surrogate labels are then used to train a PBT-S-VAE (MIG), yielding a fully disentangling VAE.

# B    PBT Training Details

We present the evolution of hyperparameters during population based training. While during supervised PBT-S-VAE training the learning rate remains relatively stable across epochs, the batch size tends to increase over time (Fig. 8a,b). Moreover, we see that our training automatically discovers to anneal $\beta$ during training. We show the increase of average populations scores over epochs on supervised PBT-S-VAE runs (Fig. 9) as well as the unsupervised PBT-U-VAE `leaf-runs` (Fig. 10). The hyperparameter schedules during unsupervised PBT-U-VAE `leaf-runs` appear to be much more non-linear (Fig. 11). In this case the learning rate increases over epochs, while the batch size decreases. $\beta$ also seems to anneal overall, but with much stronger fluctuations, particularly in epoch 0.

a) Learning rate over epochs      b) Batch size over epochs      c) Beta over epochs

Figure 8: PBT uses dynamic training schedules during supervised training runs on the *dsprites* (top) as well as *shapes3d* (bottom) datasets. Displayed are the hyperparameters of the final best-performing model of each PBT run. It discovers annealing of $\beta$ over epochs (c) for both datasets. In case of *dsprites*, all trainings exhibit a decaying learning rate schedule (a), even though for MIG based training, the learning rate is substantially lower in general. Interestingly when training with DCI, the learning rate increases first before decaying. Batch size (b) has a clear upward trend toward the latter half of training and seems to trend opposite to the annealing $\beta$ (c).

a) *dsprites* scores over epochs      a) *shapes3D* scores over epochs

Figure 9: Average population score development during supervised PBT based training. During *dsprites* training (a), DCI based runs start to plateau after 10 epochs, while MIG scores runs are still improving after 26 epochs, indicating they could benefit from longer training. For *shapes3d* training, models start to converge after the 10th epoch.

Figure 10: UDR development during rPU-VAE training on *dsprites*. From left to right recursive `metaEpochs` 0 to 4 are depicted. Each score of a model of the population is shown as a circle and the maximum UDR of the population as a solid line. The later `metaEpochs` take longer to converge, since the data for training is becoming smaller and noisier with each `metaEpoch`.

Figure 11: Hyperparameter schedules during `metaEpochs` on *dsprites*. a) There is a trend of learning rate increase with each `metaEpoch` . b) Contrary to the PBT-S-VAE runs, the batch size stays relatively small, since also the dataset size is decreased with each `metaEpoch`. c) After the first `metaEpoch`, $\beta$ is drastically decreased and stays relatively low (except a couple of spikes) during subsequent `metaEpochs`.

# C  Visualization of Full Traversals

Figure 12: Full traversal of a representative rPU-VAE model trained on *shapes3d*. Latent representations (from left to right): object scale, floor color, floor color, view angle, object shape, object color, object color, inactive, wall color, wall color.

Figure 13: Full traversal of a representative rPU-VAE model trained on *celebA*. Latent representations (from left to right): skin color, inactive, background color, head rotation, inactive, hair orientation, haircut, bangs, baldness, inactive.

a) supervised *dsprites* traversals          b) unsupervised *dsprites* traversals

Figure 14: Full traversal of a representative models trained on *dsprites*. a) For comparison to the following unsupervised model, we present traversals of supervised model PBT-S-VAE trained on *dsprites*, interestingly only scale (z1), x-position(z2) and y-position are learned (z6). a) Traversals of a representative rPU-VAE model trained on *dsprites*. Again, only scale (z6), x-position(z1) and y-position are learned (z8).

# D    Datasets

| Dataset | Ground Truth Factors |
|---------|----------------------|
| Dsprites | 32xXposition, 32xYposition, 6xScale, 40xRotation, 3xShape |
| Shapes3D | 10xFloorColor, 10xWallColor, 10xObjectColor, 8xObjectSize, 4xObjectType, 15xAzimuth |
| CelebA | Shadow, Arch. Eyebrows, Attractive, Bags un. Eyes, Bald, Bangs, Big Lips, Big Nose, Black Hair, Blond Hair, Blurry, Brown Hair, Bushy Eyebrows, Chubby, Eyeglasses, Goatee, Gray Hair, Heavy Makeup, High Cheekbones, Male |

Table 1: Summary of Ground Truth Factors for the *dsprites*, *shapes3d* and *celebA* datasets

# E Hyperparameters

## E.1 Architecture and Parameters of $\beta$-TCVAE

| Encoder | Decoder |
|---|---|
| Input: [64,64,num channels] | FC, 256 ReLU |
| 4x4 conv, 2 strides, 32 ReLU | FC, 4x4x64 ReLU |
| 4x4 conv, 2 strides, 32 ReLU | 4x4 upconv, 2 strides, 64 ReLU |
| 4x4 conv, 2 strides, 64 ReLU | 4x4 upconv, 2 strides, 64 ReLU |
| 4x4 conv, 2 strides, 64 ReLU | 4x4 upconv, 2 strides, 64 ReLU |
| FC, 2 strides, 64 ReLU | 4x4 upconv, 2 strides, num channels |

Table 2: VAE-architecture used in this study for PBT-U-VAE (UDR) and PBT-S-VAE (MIG)

## E.2 PBT hyperparameters

In our approach, unless defined otherwise, we implement the algorithm in the following way: A member $\mathcal{M}$ consists of a $\beta$-TCVAE (Table 2) with parameters $\theta$ and optimizable hyperparameters $h = \{learning\ rate, batch\ size, \beta\}$. $step$ is one epoch of SGD using Adam over the entire dataset. In $exploit$, the bottom 20% of the members according to their $p$ each randomly copy the parameters $\theta$ of one of the top 20% of the members, $explore$ perturbs the hyperparameters $h$ by a factor randomly chosen from $\{0.5, 0.8, 1.2, 2\}$.

## E.3 rPU-VAE Hyperparameters

| Parameter | Value |
|---|---|
| $UDR_{threshold}$ | 0.1 |
| $threshold_{UDR}$ | 0.005 |
| $UDR_{patience}$ | 5 epochs |
| $UDR$ num models | 5 |
| $dataset - size_{min}$ | 10 |
| PBT population size | 56 |
| PBT parameters | [$\beta$, learning rate, batch size] |
| PBT pertubation factors | [2, 1.2, 0.8, 0.5] |
| PBT init batch sizes | [8, 16, 32, 64, 128, 256, 512, 1024] |
| PBT init learning rates | $(10^{-5}...10^{0}, num = 30)$ |
| PBT init $\beta$ | $(1.5^{1}...1.5^{15}, num = 24)$ |
| rPU-VAE supervised epochs | 16 |
| rPU-VAE supervised scoring function | MIG |
| VAE z dimension | 10 |
| number of `leaf-runs` | [0,1,2,3] |
| $threshold_{z_{active}}$ | 0.5-1.0 |

Table 3: rPU-VAE Hyperparameters