[Reviews · NeurIPS 2020]

Review 1

Summary and Contributions: This work provides a novel disentanglement method by recursively learning to account for different variations in the dataset.

Strengths: - The intuition and implementation of the recursive strategy seem sound. - Figure 5 is very convincing that different factors are indeed learned sequentially - Empirical results (such as Figure 4) are convincing, with the low variance quite desirable.

Weaknesses: - In Figure 2, the size of the sprite often varies along with the xy position, which suggests lack of disentanglement to me. - In Figure 1, the right panel has a method simply labeled [14] with no name. - S2.2, what kind of object "eval" and "step" are is never explicitly defined. - The font of Figure 4 is very hard to read - While I know that the multi-image panels such as in Figure 5 are common for this field, these are a bit small to see accurately: even digitally but especially in print. I would suggest reducing the number of images, or stacking the pajnels vertically to increase the width. Very Minor/Finnicky: L152, present tense is acceptable for describing experiments, but "We now test" is a bit too narrative.

Correctness: To the best of my knowledge, but I am not very knowledgeable about disentanglement. From a high-level standpoint, the concept of iteratively reducing a problem is akin to some fundamental methods in linear algebra or other fields. Since the classification of the factors is a well-posed problem given the deterministic nature of the dataset, that implies that with s

Clarity: Overall yes. Despite my unfamiliarity with the field, it was quite easy to follow. That said, I think that neither the text and visual presentations are well-polished. The authors communicate well, but the presentation is relatively lacking. Specifically, figures are fairly consistently undersized and code variables in the text are often confusing.

Relation to Prior Work: Yes. I believe S2 provides a good overview. I would suggest a short paragraph on similar recursion-based methods. Gram-Schmidt is a classical example that is very analogous .

Reproducibility: Yes

Additional Feedback: Capitalization on section headers is inconsistent In Figure 4 (particularly 4b), the skew of the distributions seems consistent across other methods, but markedly different for your methods (Besides just low variance). Do you have any insight for why this might be? To summarize: I think that this paper does seem to present a new SotA and explains it well. The communication is effective, but not well-polished from a visual standpoint. The method is fairly complex, but the intuition is simple. =============== Update after rebuttal: My concerns re: presentation were addressed as expected; it's hard for an author to fully assuage concerns like that in a 1-page rebuttal to multiple reviewers. Seeing other reviews and discussion had made me appreciate the results and method even more. I'm tentative in my assessment due to my unfamiliarity with the field but I'm increasing my rating up to a 7 and recommending acceptance for this paper.


Review 2

Summary and Contributions: **** After author feedback **** Having read the other reviews and the author feedback, I feel very happy about the paper. I am going to keep my score but will also happily argue for the paper to be accepted. ****** This paper proposes a novel approach for robust training of disentangled VAE-based approaches in an unsupervised manner, thus addressing one of the biggest problems with the existing methods for unsupervised disentangled representation learning. The approach involves a recurrent process which trains beta-TCVAE models to disentangle a subset of dimensions at a time. The discovered dimensions are then removed from the dataset through an automatic process, and another set of beta-TCVAE models are trained to discover some of the remaining disentangled dimensions. This process is repeated until all disentangled dimensions are discovered. Each iteration uses PBT for hyper parameter tuning, combined with the recently proposed unsupervised UDR score for model evaluation.

Strengths: The authors are able to demonstrate convincing results on two datasets (dSprites and Shapes3D), achieving high disentanglement and low variance in final model performances. The paper is very well written and systematically evaluates how each step of the proposed algorithm performs. The main strength of this paper IMO is that it proposes a pipeline that addresses one of the biggest shortcomings of unsupervised VAE-based disentangling approaches - that of high variability in the quality of the learnt representation even when trained with the same hyperparameters, and the fact that these approaches often tend to learn only a subset of the generative facts. These two limitations create a problem for applying unsupervised disentangled representation learning in practice. By proposing an automated pipleline that removes these limitations, this paper is likely to pave way for a large body of applied disentanglement work.

Weaknesses: The weakness of this paper is that it doesn't necessarily create any theoretical contributions to the field. It describes a highly engineered approach instead, and this approach relies on a lot of computing power (due to relying to training numerous models through PBT+UDR). This, however, should not affect the chances of this paper of being accepted to NeurIPS, since I believe it will still be of high interest to a wide range of practitioners interested in applying disentangled representation learning to their work.

Correctness: Yes

Clarity: Yes

Relation to Prior Work: Yes

Reproducibility: Yes

Additional Feedback: Given that this paper is primarily engineering/experimental, I would have liked to see additional results to verify the success of the proposed pipeline: 1. Applying this method to more complex datasets that don't necessarily have good labels for semi-supervised disentanglement (at least CelebA) 3. Latent traversals for the final models trained to disentangle all the generative factors (on dSprites and Shapes3D, as well as the more complex dataset suggested above).


Review 3

Summary and Contributions: The paper explores using population-based training (PBT) for improving disentangled VAEs. The paper shows that applying PBT with (semi-)supervised evaluation function significantly reduces the variance of the results. In addition, the paper proposes an unsupervised approach by incorporating UDR as the evaluation function. To further improve the result, the paper proposes rPU-VAE which recursively labels the dataset with UDR and finally uses the labels to conduct another iteration of PBT-S-VAE. The result shows that this unsupervised training approach achieves better disentanglement performance than some baselines.

Strengths: Although PBT is not a new method, demonstrating the effectiveness of PBT on training disentangled VAEs is still valuable. The most interesting part is the proposed recursive factor removing algorithm. Beyond the evaluation in the paper, I can see that this idea might be useful for disentangling more challenging datasets (e.g. celeba) where it is hard to disentangle all factors by one training pass.

Weaknesses: I have some doubts about the evaluation of the (semi-)supervised part. The presentation of some important details is also obscure. This makes it hard for me to judge the soundness of the algorithm and evaluation. Please explain these details in the rebuttal and I'll adjust my score accordingly. - Section 3.2: I don't fully understand how you train "supervised" and "semi-supervised" PBT-S-VAE. Do you mean that in the supervised setting, you use *all* labeled images to evaluate MIG, whereas in the semi-supervised setting, you use only 1000 labeled images to evaluate MIG? If that's the case, I think the comparison in Figure 1 is unfair and insufficient, because beta-TCVAE doesn't use any labeled data, but (1000-)PBT-S-VAE uses. Besides comparing with [15], a more natural baseline would be to train a set of beta-TCVAE models and then use the labeled datasets (either the entire dataset or 1000 samples) to do a supervised model selection, as PBT implicitly does this model selection as well. The score of the beta-TCVAE with supervised model selection will definitely be higher, and I would imagine that the variance will also be much smaller. I am not sure how much benefit PBT has in this (semi-)supervised setting. - Line 113: "beta-VAE metric as well as the FactorVAE require the true underlying generative model and are therefore not suited". What do you mean by "underlying generative model"? If you mean a set of images with ground truth labels, I think MIG and DCI also need that. In particular, the publicly released dSprites dataset contains all the information you need for computing MIG, DCI, beta-VAE, and FactorVAE score. - The description around line 240 about leaf-run is hard to understand. From the description, it seems like in each metaEpoch, you do multiple leaf-runs. But Algorithm 1 shows that leaf-run is in the outer loop. I don't understand what leaf-run is and what's its purpose. Also, how do you intersect the intervals of z_active? Which intervals are intersected? What is "n"? How does each metaEpoch generate more surrogate labels as you are doing argmax in line 207? - Figure 3: what's the meaning of latent-encoding-value? I don't think the latent codes of VAEs can be such large. Minor issues: - Line 130: what do you mean by "upper limit"? - Figure 1(b): [14] -> [15]

Correctness: Generally yes.

Clarity: Generally yes.

Relation to Prior Work: Yes.

Reproducibility: Yes

Additional Feedback: ----- Thank the authors for the detailed response! The response answers my questions clearly, and therefore I increase the score. However, I would recommend the authors to polish the writing regarding these questions in the revison. Especially, the intention of (semi-)supervised experiments should be highlighted better. The current writing and experimental comparisons are misleading. ----- I suggest the author clarify the above points in the rebuttal.


Review 4

Summary and Contributions: In this paper, the authors propose population based training for variational training with a goal of achieving more consistent disentanglement of factors. To validate the effectiveness of the proposed method, the authors compare the extent of disentanglement for different datasets and demonstrate improved performance over the baselines. ================================================================Update after rebuttal========================================================== In my initial review, I felt the work is incremental because it is a direct application of population based training to variational training and the authors failed to provide enough evidence supporting stability of the proposed approach and the evaluation was mostly on simple datasets. After reading the author response, I am convinced of the effectiveness and the stability of the proposed method based on the model performance on CelebA datasets provided in the author response. Additionally, the authors try to address the questions as much as they could in a one page rebuttal. Based on the author response, I would like the paper to get accepted (trusting that the authors would make clarity based revisions promised in the author response). Therefore, I have increased my score from 4 to 6.

Strengths: - The paper presents the idea of population based training for VAE in multiple settings, i.e., supervised, semi-supervised, and unsupervised. The evolution of the model and highlighting the challenges and effectiveness of each settings is interesting. - The empirical evaluation highlights the potential of population based training for disentanglement in unsupervised learning.

Weaknesses: I think the contribution of the work is incremental. The model simply applies population based training to variational training. One interesting aspect of the paper is to address hyperparameter sensitivity issue. While the authors also claim hyperparameter sensitivity as one of the goals that they wish to achieve, the paper fails to provide thorough analysis and study of this aspect. The authors only briefly highlight of in Supplementary but I would like to see more analysis. These are the two main reasons I think the paper falls short of NeurIPS standards. Details oriented questions: - The motivation of rPU-VAE, i.e., recurrent version of the model in unsupervised setting is not entirely clear to me. Given that the major improvement is in the first run, what is the model achieving in the future runs? - The model description also lacks explanation on what is formal definition of active latent factors from each run? Are they determined based on variations in the latent factors? Also, discussion in the results highlighting what the model actually learns as active latent factors. - Typo: plot in Figure 1(b): [14] -> [15]

Correctness: The claims and method are technically sound.

Clarity: I think the paper needs some more clarifications on the various variants of the population based training models presented in the paper.

Relation to Prior Work: Prior work has been discussed clearly.

Reproducibility: Yes

Additional Feedback:

[Author Response · NeurIPS 2020]

We thank all the reviewers for the valuable and thorough feedback. We appreci-
ate the recognition of *"addressing one of the biggest problems with the existing
methods for unsupervised disentangled representation learning"* by [**R2**] with
the most expertise. Our results beat benchmarks [**R1**, **R2**, **R3**] by roughly dou-
bling the disentanglement scores while more than halving the variance across
all results [**R2**, **R3**] and therefore sets a new state-of-the-art for the field [**R1**]
and is *"likely to pave way for a large body of applied disentanglement work."*
as [**R2**] puts it. We respond to selected comments below but will address all
concerns in the final version.

[**R2**, **R3**] **CelebA:** We recognise that the demonstration of our method on
*celebA*, beyond the benchmarks, would add some value. We therefore in-
clude preliminary results, particularly traversals of an exemplary model on
*celebA* (Figure 1). $z_{active}$ seem to disentangle the following: **z1**-Skin Color,
**z3**-Background Color, **z4**-Head Rotation, **z6**-Hair Orientation, **z7**-Haircut, **z8**-
Bangs, **z9**-Baldness. In the final submission we will include results from all
models trained on *celebA*. [**R1**] **Remark Figure 2:** Indeed, this figure illus-
trates the consistency across models, which all learn to disentangle in the first
`metaEpoch` the x,y position. Only in subsequent `metaEpochs` the other factors
are learned. Model 1 of Figure 2 is used for the learning across `metaEpochs` in

Figure 1: qualitative evaluation of *celebA* disentanglement.

Figure 5, where one can see that in `metaEpoch` 2 scale is learned and subsequently shape/rotation in `metaEpoch` 3. [**R1**]
**Eval/step definitions:** We will add formal definitions in the final manuscript, additionally to the ones already present
in appendix E.2 (step) and Section 3.1 (eval). [**R1**] **Recursive linear algebra:** We appreciate the link to recursive
linear algebra methods like Gram-Schmidt and will discuss them in the background section. [**R1**] **Figure 4 skewness:**
One reason could be the number of models used for the violin plot. As described in the text, the reference figures
consist of 50 models, as opposed to our 5 models. In the final paper, we will overlay a swarmplot over the violinplot
to directly visualize that difference. [**R2**] **Latent Traversals:** We included full latent traversals for *shapes3d* in the
manuscript (Figure 12), we will add in the same section full traversals for *dsprites* and *celebA*. [**R3**] **(Semi-)supervised
comparison:** Your understanding of the (semi-)supervised setting is correct. The comparison with $\beta$-TCVAE is not
fair, its intention is to serve as an example of vanilla unsupervised performance. In the (semi-)supervised setting, we
mainly validate our approach and set an **upper limit** for the realistically achievable performance in the unsupervised
setting, as the model trained with the surrogate labels cannot be better than models trained with the real labels. We
nevertheless agree that the model selection $\beta$-TCVAE is a better baseline and will include it in Figure 1b) of the final
submission. Preliminary results indicate that 1000-PBT-S-VAE still consistently outperforms them. [**R3**] **Underlying
factors:** For computing the MIG and DCI scores, having images and labels is sufficient. Contrary to that, the FactorVAE
metric as well as $\beta$-VAE metric require to sample from the generative model and perform interventions (Sec. 3.1 in
[15]). For instance fixing one factor and varying all others randomly is used to estimate empirical variation in each
dimension, in case of FactorVAE, similarly for the $\beta$-VAE (Sec. 4 in [23]). [**R3**, **R4**] **Description of `leaf-runs`:**
Each `leaf-run` consists of multiple `metaEpochs`, where after each `metaEpoch`, the learned factors are removed from
the dataset and training. Think of the intersection of $z_{active}$ intervals as the set intersection of the sets of all images
where the respective latent gets mapped onto a given interval. During each `leaf-run`, a subset of the original dataset
gets surrogate labels, increasing the numbers of `leaf-runs` therefore increases the amount of total surrogate labels.
In a complete dataset, we could recursively generate a tree from all $z_{active}$ intervals, where the height of the tree is
the number of all learned factors. For efficiency reasons, we restrict the labeling to `leaf-runs`, going from the root
directly to the leaf, without further branching out. We will clarify in the revision. [**R3**] **Figure 3 LEV**: x-axis will be
correctly labelled with 'image indices' and not latent encoding value, hence the large numbers. [**R4**] **Only PBT:** The
introduction of PBT to variational training is only the first step of our work, introduced to get robust results across
models in each pass. As pointed out by [**R1**,**R2**,**R3**] the subsequent recursive approach is really the essential part of
our work, which allows it to beat all current benchmarks in unsupervised disentanglement learning by tremendous
margins. [**R4**] **Hyperparameters:** When we refer to hyperparameter sensitivity, we are particularly referring to the
very fundamental problem with the variable outcome of VAEs across datasets as pointed out by Locatello et al. [7],
whereas our algorithms unoptimized hyperparameters worked across all datasets equally and exceptionally well. We
will expand the discussion of hyperparameter schedules in supplementary section "B PBT Training Details". [**R4**]
**Motivation for recursive model:** Models did consistently only learn x,y position in the first `metaEpoch` (Figure 3).
Only in subsequent `metaEpochs` more factors are learned. More `leaf-runs` increase the amount of surrogate labels,
therefore increasing performance and robustness significantly. The exact amount of `leaf-runs` needed will ultimately
depend on the dataset. [**R4**] **Active latents:** We'll add the following formal definition of active latent factors, following
[8]: if $KL(q(z_a|x)||p(z_a)) > 0.01$, $z_a$ is active, otherwise not. [**R4**] **Latent representations:** Supplementary C
describes the learned factors for *shapes3d*, we will add a similar figure for *dsprites*, a proxy can be seen in Figure 5. We
will add a sentence in the discussion of the final manuscript, describing these learned factors as well as *celebA* results.

[Meta-Review · NeurIPS 2020]

This paper proposes a new method for learning disentangled representations with VAEs. The main contribution is that the proposed method is robust, in the sense that training reliably converges. This addresses a shortcoming in existing methods, where the quality of the learned representation is often highly variable. The approach proposed approach involves a recurrent process which trains beta-TCVAE models to disentangle a subset of dimensions at a time. The discovered dimensions are then removed from the dataset through an automatic process, and another set of beta-TCVAE models are trained to discover some of the remaining disentangled dimensions. This process is repeated until all disentangled dimensions are discovered. Each iteration uses PBT for hyper parameter tuning, combined with the recently proposed unsupervised UDR score for model evaluation. This submission was positively received by reviewers, who found the experimental results convincing, and noted that the submission indeed addresses a key issue with existing approaches by improving robustness. Several reviewers did not issues with clarity and presentation. The authors provided a detailed response, and reviewers' opinion of the paper improved during discussion (please see updated reviews). The AC will recommend this submission for acceptance with a couple of caveats. Having looked at the submission, the AC is inclined to agree with reviewers that this submission in its current form is a little rough around the edges. Please work carefully to implement all changes that were discussed in the response, address all remaining minor comments by reviewers, and make sure that the camera ready version of this paper is impeccably presented. The AC would also like to gently correct the authors on a couple points of communication (which will help them in future borderline cases): First, please break up your author response into paragraphs. Making key points stand out is much more effective than making as many points as possible. Second, please do not use optional comments to chairs to further argue your case. The AC carefully looks at every paper, author response, and review, and will in many cases communicate with reviewers to clarify their reviews prior to the response phase. Reserve your comments for exceptional cases, e.g to request an additional review or to point out a potential form of reviewer misconduct. Finally, please be consistent in referring to all reviewers as he/she or they; inadvertent inconsistencies will be noticed and can be easily misconstrued.